

# Optically effective complex refractive index of coated black carbon aerosols: from numerical aspects

Xiaolin Zhang[1,2], Mao Mao[1,2]

[1]Key Laboratory of Meteorological Disaster, Ministry of Education (KLME) / Collaborative Innovation Center on Forecast
and Evaluation of Meteorological Disasters (CIC-FEMD) / Joint International Research Laboratory of Climate and
Environment Change (ILCEC) / Earth System Modeling Center (ESMC) / Key Laboratory for Aerosol-Cloud-Precipitation
of China Meteorological Administration, Nanjing University of Information Science & Technology, Nanjing, 210044, China
[2]School of Atmospheric Physics, Nanjing University of Information Science & Technology, Nanjing, 210044, China

*Correspondence to*: Mao Mao (mmao@nuist.edu.cn) or Xiaolin Zhang (xlnzhang@nuist.edu.cn)

**Abstract.** Aerosol complex refractive index (ACRI) is an important microphysical parameter used for the studies of
modeling their radiative effects. With considerable uncertainties related to retrieval based on observations, a numerical study
is a powerful method, if not the only one, to provide a better and more accurate understanding on retrieved optically effective
ACRI of aged BC particles. Numerical investigations on the optically effective ACRI of polydisperse coated BC aggregates
retrieved from their accurate scattering and absorption properties, which are calculated by the multiple-sphere T-matrix
method (MSTM), without overall particle shape variations during retrieval, are carried out. The aim of this study is to
evaluate the effects of aerosol microphysics, including shell/core ratio $D_p / D_c$, BC geometry, BC position inside coating,
and size distribution, on retrieved optically effective ACRI of coated BC particles. At odds with expectations, retrieved
optically effective ACRIs of coated BC particles in coarse mode, are not merely impacted by their chemical compositions
and shell/core ratio, being highly complicated functions of particle microphysics. However, in accumulation mode, the
coated BC optically effective ACRI is dominantly influenced by particle chemical compositions and shell/core ratio,
although it shows slightly sensitive to BC geometry, BC position inside coating and particle size distribution. The popular
volume weighted average (VWA) method and effective medium theory (EMT) provide acceptable ACRI results for coated
BC in accumulation mode, and the resulting uncertainties in particle scattering and absorption are both less than
approximately 10%. For coarse coated BC, the VWA and EMT, nevertheless, produce dramatically higher imaginary parts
than those of optically effective ACRIs, significantly overestimating particle absorption by a factor of nearly 2 for heavily
coated BC with a large BC fractal dimension or BC close to coating boundary. Using the VWA could introduce significant
overestimation in aged BC absorption analysis studies, and this may be one of the reasons why modelled aerosol optical
depth is 20% larger than observed, since it is widely employed in the state-of-the-art aerosol-climate models. We propose a
simple new ACRI parameterization for fully coated BC with $D_p / D_c \geq 2.0$ in coarse mode, which can serve as a guide for
the improvement of ACRI of heavily coated BC, and its scattering and absorption errors are reduced by a factor of nearly 2
compared to the VWA. Our study indicates that a reliable estimate of the radiative effects of aged BC particles in coarse





mode would require accounting for the optically effective ACRI, rather than the ACRI given by the VWA, in aerosol-climate models.

# 1 Introduction

The largest uncertainty in estimates of the effects of atmospheric aerosols on climate stems from uncertainties in the determination of their microphysical properties, which in turn determines their optical properties. As one of the most significant microphysical properties, aerosol complex refractive index (ACRI) should be known for modeling their radiative effects, and the magnitude of radiative forcing is very sensitive to the ACRI, especially the imaginary part [Raut and Chazette, 2008a]. The ACRI is determined by particle chemical composition, governing its inherent scattering and absorption properties.

Black carbon (BC), emitted from incomplete fossil fuel combustion and biomass burning, can be coated with secondary aerosol species (e.g., organics and sulfate) through the aging process, being one of the largest uncertainties in estimating aerosol radiative forcing due to their complicated geometry and mixing state [Ramanathan and Carmichael, 2008; Myhre, 2009; Bond et al., 2013; Zhang et al., 2015]. As a strong absorptive aerosol, pure BC particles have a large ACRI, whereas our understanding on the ACRI of aged BC is still limited because of its internally mixing with weakly absorptive coatings [e.g., Shiraiwa et al., 2010]. The ACRI of BC internal mixtures, named as effective ACRI, are normally obtained based on the volume weighted average (VWA) method and effective medium theory (EMT), and the choice of both approaches is driven by high dependency of ACRI on particle chemical compositions [e.g., Kandler et al., 2009]. The state-of-the-art aerosol-climate models employ the VWA method extensively, approximating the effective ACRIs of internal- and external-mixed aerosol ensemble at each mode for calculating their optical and radiative properties [e.g., Stier et al., 2005; Kim at al., 2008; Zhang et al., 2012]. Nonetheless, the performances of the VWA and EMT are open questions, as several studies have questioned the validity of the both approximations in some questions [e.g., Voshchinnikov et al., 2007].

The estimates of the ACRI of coated BC can also be made from observed optical properties, and the ACRI is inferred by obtaining a best fit to numerical simulations with Mie theory assuming spherical particle shape, which is called as optically effective ACRI. For instance, the optically effective ACRIs are retrieved based on simultaneous measurements of surface aerosol scattering and absorption coefficients, as well as size distributions [Riziq et al., 2007; Schkolnik et al., 2007; Mack et al., 2010; Stock et al., 2011]. Meanwhile, the airborne in situ measurements of particle optical properties from a Particle Soot Absorption Photometer (PSAP), Spectral Optical Absorption Photometer (SOAP), sunphotometer, or lidar, combined with a Mie theory based data analysis scheme, are also applied for the retrieval of optically effective ACRI [Raut and Chazette, 2008a, 2008b; Petzold et al. 2009; Muller et al. 2009]. Muller et al. [2010] even compare retrieved optically effective ACRIs from different techniques, and reveal only partly a reasonable agreement with significant differences for the spectra of imaginary part remaining, indicating uncertainties during retrieval. The uncertainties may be that those retrieval methods are based on unrealistic spherical shape assumption, inaccurate numerical modeling, or without considering the errors in aerosol



optical measurements, and then, sizeable errors on retrieved optically effective ACRIs are posed. Furthermore, these uncertainties significantly limit our ability to understand the relationships between the optically effective ACRI and aerosol other microphysical properties, and furthermore to improve radiation simulations in aerosol-climate models. Therefore, a systematic theoretical investigation on optically effective ACRIs of internal-mixed particles retrieved from exactly

calculated optical properties without particle shapes changed is a must, which is generally missing, and will benefit our understanding on these relationships. For coated BC particles with several chemical compositions, their optically effective ACRIs are not only affected by their compositions, but possibly impacted by their other microphysics. However, the effects of coated BC microphysics on their optically effective ACRIs are still under discussion and need more investigation.

Here, numerical investigations of the optically effective ACRIs of polydisperse coated BC aggregates as examples are

systematically presented based on our current understanding, and the optically effective ACRI influences are decomposed into that due to particle microphysical properties, including shell/core ratio, BC fractal dimension, BC position inside coating, and size distribution. An exact multiple-sphere T-matrix method (MSTM) is employed to numerically calculate the absorption properties of coated BC aggregates while the Mie method is applied for the retrieval of optically effective ACRI. The objective is to evaluate the effects of coating microphysics on the optically effective ACRIs of aged BC particles, which

hopefully benefits our understanding of the mechanism responsible for the model-observation discrepancies and refining estimates of aerosol radiative forcing. The performances of the VWA and EMT approximations are also studied for comparison.

## 2 Methodology

### 2.1 Coated BC model

Freshly emitted BC particles often exist as loose cluster-like aggregates with hundreds or even thousands of small spherical monomers [e.g., Li et al., 2016], and the concept of fractal aggregate has shown great success and wide applications to represent realistic BC geometries [e.g., Sorensen, 2001]. The fractal aggregate can be mathematically described by the well-known statistic scaling rule following

$$N = k_0 \left( \frac{R_g}{a} \right)^{D_f} , \tag{1}$$

$$R_g = \sqrt{\frac{1}{N} \sum_{i=1}^{N} r_i^2} , \tag{2}$$

where $N$ is the monomer number in an aggregate, $a$ the mean monomer radius, $k_0$ the fractal prefactor, $D_f$ the fractal dimension, and $R_g$ the gyration radius.



After being emitted into the atmosphere, BC aggregates tend be coated by other materials, such as sulfate and organics [e.g., Schwarz et al., 2008; Tritscher et al., 2011], through the aging process, and their chain-like structures tend to collapse into more compact clusters [Zhang et al., 2008; Coz and Leck, 2011]. The aged BC particles can have BC $D_f$ of almost 3, while the fresh BC aggregates generally show lacy structures with $D_f$ less than 2 [Liu et al., 2008]. While the fractal aggregates

have been successfully employed to model the geometries of BC particles [e.g., Dlugach and Mishchenko, 2015; Mishchenko et al., 2016], their coating geometries are generally complicated in ambient air. Some observations of individual aged BC particles actually show spherical coating geometry [e.g., Schnaiter et al., 2005; Alexander et al., 2008; Zhang et al., 2008; Wu et al., 2016], while others depict complex irregular geometries. Meanwhile, it is found that the simple spherical coatings on BC particles have similar effects on scattering and absorption properties to those with more complicated coating

structures [e.g., Dong et al., 2015; F. Liu et al., 2015]. To avoid the influence of overall particle shape variations on retrieval results of optically effective ACRI and use the fast Mie theory for retrieval, this study therefore considers an aggregate as the BC core, and a spherical coating is added as the coating material, which is assumed to be the weakly absorbing sulfate, following the numerical model developed by Zhang et al. [2017]. The sketch map of the BC aggregates coated by sulfate with an overall spherical shape is shown in Fig. 1.

For this inhomogeneous internal-mixed particle, the BC aggregates are generated based on a tunable particle-cluster aggregation algorithm from Skorupski et al. [2014]. The $k_f$ of BC is assumed to be 1.2 based on Sorensen. [2001]. The radius $a$ of BC aggregate monomers are observed to vary over a range of about 10–25 nm [Bond and Bergstrom, 2006], while the monomer number $N$ can alter up to approximately 800 [Adachi and Buseck, 2008]. Since the monomer size has a rather weak effect on BC scattering and absorption as $D_f$ is fixed [Liu and Mishchenko, 2007; He et al., 2015], we consider

two $N$ values of 200 and 800 as examples for accumulation and coarse particles, respectively, and compare three $D_f$ values of 2.6, 2.8 and 2.98 for aged BC aggregates. After BC geometry being defined, the shell/core ratio $D_p / D_c$ of coated BC is assumed to be in the range of 1.1–2.7 on the basis of the SP2 measurements in London [D. Liu, et al., 2015] and Beijing [Zhang et al., 2016]. It should be noted that, some small $D_p / D_c$ values might not be used, because we only study the cases for BC aggregates fully coated by sulfate. An incident wavelength of 550 nm is considered in this study, and related

refractive indices of BC and sulfate are assumed as $1.85 - 0.71i$ [Bond and Bergstrom, 2006] and $1.52 - 5.0 \times 10^{-4} i$ [Aouizerats et al., 2010], respectively. With the internal-mixed coated BC model defined, which depicts quite realistic geometries, its random-orientation scattering and absorption properties are exactly calculated with the robust multiple-sphere T-matrix method [Mackowski, 2014].

For ambient atmospheric applications, it is meaningful to consider bulk particle optical properties averaged over a certain

size distribution. This study explores an ensemble of BC aggregates with different sizes but the same sulfate coating fraction (i.e., same $D_p / D_c$), and a lognormal size distribution is assumed with the form of:





$$n(r) = \frac{1}{\sqrt{2\pi} r \ln(\sigma_g)} \exp\left[ -\left( \frac{\ln(r) - \ln(r_g)}{\sqrt{2} \ln(\sigma_g)} \right)^2 \right], \tag{3}$$

where $\sigma_g$ is the geometric standard deviation, and $r_g$ the geometric mean radius [e.g., Yurkin and Hoekstra, 2007; Schwarz et al., 2008]. As particles in accumulation and coarse modes contribute dominated light scattering and absorption, we only consider coated BC in both modes. For the accumulation mode, the radius range is set as 0.05–0.5 μm, while the coarse radius range is assumed to be 0.5–2.5 μm as ambient aerosols with size larger than 5 μm are few [Zhang et al., 2014, 2018; Zhang and Mao, 2015]. To better understand the behavior at different particle size mode, size distributions in accumulation and coarse modes are utilized separately, which are similar to those applied in the aerosol-climate models [Zhang et al., 2012]. In this study, we consider the size distributions of coated BC aggregates (i.e., BC-sulfate internal mixtures) with $r_g$ of 0.075 μm and 0.75 μm, and $\sigma_g$ of 1.59 and 2.0 in accumulation and coarse modes, respectively [Zhang et al., 2012]. With given particle size distributions, the bulk scattering cross section ($C_{sca}$) and absorption cross section ($C_{abs}$) of coated BC follow the equations of:

$$\langle C_{sca} \rangle = \int_{r_{\min}}^{r_{\max}} C_{sca}(r) n(r) d(r), \tag{4}$$

$$\langle C_{abs} \rangle = \int_{r_{\min}}^{r_{\max}} C_{abs}(r) n(r) d(r). \tag{5}$$

With the inhomogeneous coated BC model defined and its exact scattering and absorption properties obtained, it is possible to retrieve its optically effective ACRI with more details.

## 2.2 Retrieval approach

The retrieval approach is similar to the methods described in previous studies [e.g., Mack et al., 2010; Stock et al., 2011; Zhang et al., 2013], with the only differences being that the inherent aerosol optical properties are exactly calculated rather than measured and particle overall shapes are not changed during retrieval. Among all particle optical properties, the scattering and absorption are selected for retrieval, since both are basically governed by the real and imaginary part of ACRI, respectively. As coated BC models are overall spherical, the optically effective ACRI is determined by an iterative algorithm based on Mie theory, utilizing particle size distributions and calculated scattering and absorption cross sections. Exploiting all calculations, the designed inversion scheme to retrieve the optically effective ACRI follows.

Based on a guess for a real, n, and imaginary part, k, of the ACRI at a given wavelength, two look-up tables are built from the database with known size distribution. One look-up table encompasses the scattering cross sections and the other contains the absorption cross sections. For physically-based sense, guessed real and imaginary parts of the optically effective ACRI are within refractive index ranges of known compositions of simulated internal-mixed particles. Thus in our retrieval, the guessed real part of refractive index varies from 1.52 to 1.85 with an equidistant space of 0.001 and the imaginary part



changes from $5.0\times10^{-4}$ to 0.71 with a logarithmic interval of 0.005. Then the retrieval algorithm simultaneously varies n and k, and scans through all physically possible ACRI values within a selected resolution until it minimizes $\chi^2$:

$$\chi^2(n,k) = \frac{1}{N}\sum_{i=1}^{N_\lambda}\left[\left(\frac{C_{sca,calculated}(n,k)-C_{sca,inherent}}{C_{sca,inherent}}\right)_i^2 + \left(\frac{C_{abs,calculated}(n,k)-C_{abs,inherent}}{C_{abs,inherent}}\right)_i^2\right], \tag{6}$$

where $C_{sca,inherent}$ and $C_{abs,inherent}$ are inherent scattering and absorption cross sections of simulated internal-mixed particles,
$\chi^2(n,k)$ generates the fractional difference of the calculated scattering and absorption cross sections relative to the inherent properties, and $N$ is the number of calculations during the retrieval. The $\chi^2(n,k)$ values for particle scattering and absorption are minimized by optimizing initial guess ACRI values, yielding an optically effective ACRI at this wavelength. As particle scattering is mainly determined by the magnitude of n while its absorption is primarily governed by the magnitude of k, the minimization of $\chi^2(n,k)$ should retrieve a unique result of optically effective ACRI.

The optically effective ACRI of internal-mixed particles here is defined as an ACRI that provides almost the same scattering and absorption properties as their inherent properties, based on known size distribution and overall particle shapes for homogeneous particles. Please note that aged BC particles have complicated shapes in ambient air [Liu et al., 2017], coated BC considered in this study represents a case study, resembling the findings presented by Schnaiter et al. [2005], to give insights into the effects of particle microphysics on its optically effective ACRI.

## 3 Results and Discussion

### 3.1 Effect of coated BC morphologies on its optically effective ACRI

This study focuses on the influence of the microphysics of coated BC aggregates on their optically effective ACRIs, and, therefore, the properties of the microphysics are our interest. The coated BC optically effective ACRI depends not only on particle shell/core ratio (i.e., $D_p / D_c$) but also on particle morphology (i.e., the physical arrangement of BC with respect to
other components within a given particle). With sulfate coating geometry fixed and BC fully coated, we will consider two other morphological factors: BC geometry and BC position inside sulfate coating.

To show the effect of BC geometry on coated BC optically effective ACRI, the concentric core-shell structures (i.e., mass centers locating at coating center) with inside BC aggregates exhibiting fractal dimensions of 2.6, 2.8 and 2.98 are considered. Figure 2 compares retrieved optically effective ACRIs of these coated BC aggregates with different BC
geometries at different shell/core ratios, while the introduced differences of scattering and absorption cross sections are illustrated in Fig. 3. The differences are relative errors of scattering and absorption cross sections induced by retrieved effective ACRIs compared with initial inherent optical properties. The retrieved ACRIs of internal-mixed coated BC particles are optically effective, since the relative errors for both scattering and absorption cross sections are within 1% (see



Fig. 3a and 3b). For comparison, the ACRIs of coated BC aggregates derived from the popular volume weighted average method and Bruggeman effective medium theory, as well as their induced differences of scattering and absorption are also shown in Figures 2 and 3. The properties are averaged over an ensemble of BC-sulfate internal-mixed particles with the aforementioned size distributions for accumulation and coarse modes separately.

As shown in Fig. 2, in accumulation mode, it is expected that, as $D_p / D_c$ increases (i.e., BC content decreases), both n and k of retrieved optically effective ACRI of concentric coated BC aggregates decrease. With $D_p / D_c$ varying from 2.7 to 1.5, the real part of optically effective ACRI of accumulation coated BC increases from ～1.53 to ～1.60, whilst its imaginary part becomes almost 6 times larger from ～0.038 to ～0.216 (see Fig. 2a and 2c). As BC fractal dimension increases (i.e., BC becomes more compact), the real part of optically effective ACRI shows a slight decrease in accumulation mode,

whereas the reverse is true for retrieved imaginary part. For mass-center positions of different BC geometries fixed within sulfate coating, retrieved optically effective ACRIs of accumulation particles are slightly sensitive to their inside BC geometries, with differences less than 1% and 5% for n and k, respectively. Compared to the optically effective ACRIs of concentric coated BC aggregates in accumulation mode, the imaginary parts of ACRIs estimated from the VWA and EMT are lower by 2–7% and 9–15%, respectively, depending on $D_p / D_c$ and BC fractal dimension, while their real parts are

slightly higher with differences within 2%. As a result, the VWA and EMT overestimate coated BC scattering cross sections by 2–7% and 4–11%, and underestimate absorption cross sections by 1–4% and 5–11%, respectively (see Fig. 3c and 3e). Meanwhile, the VWA performs slightly better than the EMT for coated BC in accumulation mode.

Unlike accumulation mode, retrieved optically effective ACRI of concentric coated BC in coarse mode depicts distinctive patterns, which is illustrated in Fig. 2b and 2d. The impact of particle microphysics on the optically effective ACRIs of

coarse concentric coated BC is complicated, especially for their real parts, which show strong oscillations as a function of shell/core ratio. The imaginary parts of retrieved optically effective ACRIs of coated BC aggregates in coarse mode generally decrease with the increase of $D_p / D_c$ or BC fractal dimension. The imaginary parts of derived ACRIs based on the VWA and EMT can be higher than those of retrieved optically effective ACRIs by a factor of ～3, and resulting overestimation of absorption cross sections of coarse concentric coated BC can be as high as ～75%. The optically effective

ACRI, producing coated BC scattering and absorption with differences less than 1%, performs predominantly better than the VWA and EMT in coarse mode, and the VWA and EMT result in more uncertainties in particle absorption than scattering. Furthermore, the VWA and EMT overestimate more absorption and underestimate more scattering for coarse coated BC with larger BC fractal dimension or $D_p / D_c$.

The simulations discoursed above assume coated BC with a concentric core-shell structure, which does not always represent

realistic aerosols, whereas coated BC with an off-center core-shell structure may be certainly true for some ambient particles. Figure 4 portrays retrieved optically effective ACRIs of coated BC aggregates (BC fractal dimension of 2.8) with the aforementioned size distributions for two different off-center structures compared to the concentric core-shell structure. For




two off-center core-shell structures assumed, one is BC aggregates locating at the middle of a radius of coating sphere, and the other is BC at an outer position as close as possible to coating boundary. It is evident that coated BC optically effective ACRIs in accumulation mode decrease with increasing $D_p / D_c$ for various BC inside positions (see Fig. 4a and 4c). The optically effective ACRIs of accumulation coated BC aggregates are generally sensitive to the BC position inside sulfate

coating, with variations of 1% and 20% for n and k, respectively. When BC aggregates move from coating center to the boundary, the real parts of retrieved optically effective ACRIs in accumulation mode are found to increase slightly, as opposed to the decrease of their imaginary parts. For accumulation BC aggregates with different core-shell structures, the VWA and EMT give relatively close ACRIs to their optically effective ACRIs with n differences both within 1% and k differences less than 11% and 14%, respectively. In coarse mode, the real parts of retrieved optically ACRIs show intricately

strong variations on shell/core ratio and BC inside position, and their imaginary parts are generally decreasing with BC becoming closer to coating boundary (see Fig. 4b and 4d). The imaginary parts of retrieved optically effective ACRIs of coarse coated BC with different BC inside positions are significantly lower than those given by the VWA and EMT, indicating severe overestimation of coarse particle absorption by the VWA and EMT.

Figure 5 illustrates the differences of scattering and absorption cross sections of coated BC aggregates with different BC

inside positions induced by the VWA, EMT and optically effective ACRIs. The optically effective ACRIs cause differences of coated BC scattering and absorption within 1% compared to its inherent properties in both accumulation and coarse modes, whereas the VWA and EMT induce large particle scattering and absorption differences, especially in coarse mode. One can see that, in coarse mode, the VWA and EMT can overestimate coated BC absorption as high as ∼90% at some coating states, and they overestimate more for BC closer to coating boundary.

Generally, retrieved optically effective ACRIs of coated BC aggregates show significantly distinctive patterns in accumulation and coarse modes. In accumulation mode, besides shell/core ratio, the optically effective ACRIs are slightly sensitive to BC geometry and BC position inside sulfate coating. Their retrieved real parts increase slightly as BC becomes loose or BC locates closer to coating boundary, whereas the reverse is true for their imaginary parts. Nevertheless, in coarse mode, the real parts of optically effective ACRIs are highly complex functions of shell/core ratio, BC fractal dimension and

BC inside position, while their imaginary parts generally increase with BC fractal dimension decreasing or BC close to coating center. Meanwhile, the VWA and EMT show acceptable performance for estimating ACRIs of coated BC in accumulation mode, resulting in uncertainties of scattering and absorption both within approximately 10%. Whereas in coarse mode, the VWA and EMT, generating dramatically higher imaginary parts than those of optically effective ACRIs, can significantly overestimate coated BC absorption by a factor of nearly 2, particularly for heavily coated BC with a large

BC fractal dimension or BC close to coating boundary.



### 3.2 Effect of coated BC size distribution on its optically effective ACRI

Figure 6 illustrates the variations of retrieved ACRIs of concentric coated BC aggregates (BC fractal dimension of 2.8) with different particle size distributions at different shell/core ratios. The real parts (top row) and imaginary parts (bottom row) of retrieved ACRIs are depicted in Fig. 6, respectively, and the panels from left to right correspond to the optically effective

ACRIs in accumulation and coarse modes, and the ACRIs given by the VWA and EMT. The lognormal size distributions are assumed for the coated BC particles with $r_g$ (x axis) ranging from 0.025–0.15 μm and 0.5–1.0 μm in accumulation and coarse modes, respectively, and $\sigma_g$ fixed as the aforementioned values. Figure 6 clearly depicts that the optically effective ACRIs of coated BC aggregates are sensitive to particle size distribution and shell/core ratio. For accumulation coated BC, retrieved optically effective ACRI shows weak variation on particle size distribution, and with $r_g$ increasing, its imaginary

part increases mildly for thin coating (i.e., small $D_p / D_c$) whereas the real part decreases. Nevertheless, in coarse mode, the variation of retrieved optically effective ACRI becomes strong, and its imaginary part generally show a decrease trend as $r_g$ increases. Compared to the results in accumulation mode, retrieved optically effective ACRI of concentric coated BC aggregates in coarse mode becomes more sensitive to particle size distribution and shell-core ratio, i.e., showing larger variation. Considering that BC/sulfate volume ratio is a constant as $D_p / D_c$ is fixed, ACRI results given by the VWA and

EMT are not sensitive to particle size distribution and are expressed by the horizontal lines in the figure. In accumulation mode, the VWA and EMT provide acceptable ACRI results for different size distributions with scattering and absorption uncertainties within ～10% (not shown), and the VWA shows mildly better performance than the EMT in estimating the imaginary part. However, in coarse mode, the imaginary parts are severely overestimated by the VWA and EMT at different size distributions when compared to retrieved optically effective ACRIs, indicating that the VWA and EMT overrate aged

BC absorption significantly.

### 3.3 A new assumed ACRI parameterization for heavily coated BC

As discussed above, the VWA approximation employed in the state-of-the-art aerosol-climate models extensively, could result in significant errors in the absorption of thickly coated BC aggregates in coarse mode (specifically, $D_p / D_c \geq 2.0$), although it gives acceptable results for fully coated BC in accumulation mode and thinly coated BC in coarse mode. For

thickly coated BC in coarse mode, with all previous microphysical factors considered, it seemingly becomes possible to decompose the influences of coated BC microphysics on their optically effective ACRI and do the parameterization. Nonetheless, the optically effective ACRI of coarse coated BC are highly intricate functions of their microphysical properties parameterized by multiple parameters (e.g., BC fractal parameters, coating parameters and shell/core ratio), and cannot simply represented by these microphysical parameters adequately. Because the ACRI of internal-mixed particles is

dependent on their chemical compositions, and the traditional VWA approach to their determination is to calculate them

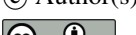



from their bulk chemical compositions and known values of the refractive indices of the pure components. This may be a reasonable approach for approximation of the real part of ACRI, whereas the corresponding values for calculation of the imaginary part are not as well as their real counterparts (e.g., Marley et al., 2001). Meanwhile, for heavily coated BC in coarse mode with various microphysics, the VWA generates dramatically higher imaginary parts than those of optically

effective ACRIs to varying degrees. Thus, the imaginary part of heavily coated BC in coarse mode is considered as that given by the VWA divided by a factor, which may show better results, whilst its real part is the same as that approximated by the VWA. To be simple and specific, the factor for dividing the imaginary part given by the VWA is assumed to be 2, and a new assumed parameterization of ACRI of BC heavily coated by sulfate in coarse mode is expressed by

$$n_{effective}(\lambda) = (n_{BC}(\lambda) \times \frac{V_{BC}}{V_{BC} + V_{sulfate}} + n_{sulfate}(\lambda) \times \frac{V_{sulfate}}{V_{BC} + V_{sulfate}}) / 2, \qquad (7)$$

$$k_{effective}(\lambda) = \frac{1}{2} \times (k_{BC}(\lambda) \times \frac{V_{BC}}{V_{BC} + V_{sulfate}} + k_{sulfate}(\lambda) \times \frac{V_{sulfate}}{V_{BC} + V_{sulfate}}) / 2, \qquad (8)$$

where $n_{effective}$ and $k_{effective}$ denote the real part and imaginary part of parameterized ACRI, $\lambda$ the wavelength of light, $n_{BC}$ and $n_{sulfate}$ the real parts of pure BC and sulfate, $k_{BC}$ and $k_{sulfate}$ the imaginary parts of BC and sulfate, and $V_{BC}$ and $V_{sulfate}$ the volumes of BC and sulfate, respectively.

To demonstrate the performance of the simple expresses on approximating ACRI of fully coated BC aggregates in coarse

mode with $D_p / D_c \geq 2.0$, Fig. 7 compares induced relative errors of scattering and absorption coefficients of coarse coated BC based on ACRIs from the popular VWA and Equations (7-8). The cases of coarse coated BC with BC fractal dimension of 2.8 and 2.98 are illustrated in left four and right four panels of the figure. It is clear that assumed new ACRI parameterization method shows a better performance than the VWA in the estimation of scattering and absorption of heavily coated BC with various coating microphysics in coarse mode. Compared to the VWA, the new parameterization method

reduces the relative errors in estimating absorption cross sections of coarse coated BC by a factor of nearly 2. Surprisingly, the errors of scattering cross sections of coated BC are also lessened, although the real part of ACRI in assumed parameterization method is considered to the same as that in the VWA. As the effects of particle microphysics on optically effective ACRI of coarse coated BC are rather complicated, it is difficult to find a "best" parameterization for the optically effective ACRI based on its microphysics. However, Fig. 7 indicates that the simple ACRI approximation we assume give a

better estimation than the VWA, which are widely used in aerosol-climate models, on the optical properties of heavily coated BC in coarse mode.



### 3.4 Atmospheric implications

Our theoretical analysis depicts retrieved optically effective ACRI of coated BC sensitive to its shell/core ratio, BC geometry, BC position inside coating, and size distribution. Due to aged BC particles having complicated coating morphologies in ambient air [Liu et al., 2017], coated BC considered in this study represents case studies, such as those BC particles observed

under polluted urban environments [Peng et al., 2016; Chen et al., 2017], to give insights into the effects of particle microphysics, resembling the findings presented by Schnaiter et al. [2005]. The study indicates that retrieved optically effective ACRIs of coated BC aggregates show distinctive patterns in accumulation and coarse modes. In accumulation mode, retrieved optically effective ACRIs of coated BC are more impacted by their chemical compositions and composition ratio, which look like the real ACRI, and the influences by their other microphysics are generally limited. However, in

coarse mode, the results challenge conventional beliefs, and retrieved optically effective ACRIs of coated BC are highly complicated functions of particle microphysics. That is to say, the optically effective ACRIs of coarse coated BC are not only affected by their chemical compositions and composition ratio, but also impacted by their other microphysics (such as size distribution and BC geometry). This makes the notion of refractive index become somewhat ill-defined for internal-mixed particles, since it is not the real particle refractive index but optically effective refractive index. This study also

indicates that the VWA and EMT, giving acceptable ACRIs for internal-mixed particles in accumulation mode, produce higher imaginary part of ACRI than that of optically effective ACRI, and could overestimate heavily coated BC absorption significantly in coarse mode. This may be one of the reasons why modelled aerosol optical depth is 20% larger than observed [Roelofs et al., 2010], since the VWA approximation is widely employed in the state-of-the-art aerosol-climate models. To reduce the uncertainties, we propose a simple ACRI parameterization method for heavily coated BC in coarse

mode (specifically, $D_p / D_c \geq 2.0$), which reduces the scattering and absorption errors of coated BC by a factor of nearly 2 in comparison with the VWA, although absorption errors of coated BC with some microphysics are still relatively high. As such, in order to produce reliable estimates of BC radiative forcing in aerosol-climate models, the optically effective ACRI, rather than the ACRI given by the VWA, appears to be essential, especially for aged BC in coarse mode.

### 4 Conclusions

This study numerically explores the impacts of coating microphysics on the optically effective ACRIs of polydisperse coated BC particles, which are retrieved from exactly calculated scattering and absorption properties without variations of overall particle shapes during retrieval. The numerical simulations conducted here have multiple controllable microphysical variables, i.e., shell/core ratio, BC geometry, BC position inside sulfate coating and size distribution, and we attempt to constrain these variables within realistic ranges as determined by observation-based studies. The fractal aggregate is

employed to model the realistic BC geometry, and optical properties of spherical coated BC aggregates are calculated utilizing the numerically exact MSTM. The fast Mie theory based data analysis scheme is applied for retrieving the optically





effective ACRIs of coated BC, and the numerical results are analyzed to better understand retrieved optically effective ACRIs in relation to the controllable microphysical variables.

Our results reveal that retrieved optically effective ACRIs of coated BC aggregates depict significantly different patterns in accumulation and coarse modes. With BC becoming loose or close to coating boundary, the real parts of retrieved optically effective ACRIs of accumulation coated BC increase slightly, as opposed to the decrease for the imaginary parts. The retrieved optically effective ACRI of coated BC in accumulation mode are predominantly influenced by their chemical compositions and composition ratio, which makes it reasonable and looks like the real ACRI, although it shows slightly sensitive to BC geometry, BC position inside coating and particle size distribution. Nonetheless, retrieved optically effective ACRIs of coarse coated BC are highly complicated functions of particle microphysics, and this challenges conventional beliefs given by the VWA and EMT. The VWA and EMT exhibit acceptable performances for estimating ACRIs of coated BC in accumulation mode, and resulting uncertainties in scattering and absorption are both within approximately 10%. In coarse mode, the VWA and EMT, nevertheless, produce dramatically higher imaginary parts than those of optically effective ACRIs, and can significantly overestimate coated BC absorption by a factor of nearly 2, especially for heavily coated BC with a large BC fractal dimension or BC close to coating boundary. This is probably one of the reasons why modelled aerosol optical depth is 20% larger than observed [Roelofs et al., 2010], as the VWA approximation is widely employed in the state-of-the-art aerosol-climate models.

Although the parameterization of the optically effective ACRI of coarse coated BC is difficult and challenging, we propose a simple ACRI parameterization method for heavily coated BC with $D_p / D_c \geq 2.0$ in coarse mode, and its scattering and absorption errors are decreased by a factor of nearly 2 compared to the VWA. Overall, this work clearly highlights the importance of accounting for the optically effective ACRI, rather than the ACRI given by the VWA, for producing reliable estimates of radiative forcing of coated BC, especially in coarse mode, in aerosol-climate models. However, caution may be taken in interpreting our results as a comprehensive guide, as the closure studies between observation and numerical models on aged BC properties are still comparatively poor [Radney et al., 2014].

*Author contribution.* X. Zhang and M. Mao designed the research plan. X. Zhang carried it out, performed the simulations, and prepared the manuscript with contributions from all co-authors.

*Acknowledgements.* We particularly acknowledge the source of the codes of MSTM 3.0 from Daniel W. Mackowski and Michael I. Mishchekno. This work is financially supported by the National Natural Science Foundation of China (NSFC) (Nos. 41505127, 21406189), Natural Science Foundation of Jiangsu Province (No. BK20150901), Natural Science Foundation of the Jiangsu Higher Education Institutions of China (No. 15KJB170009), and Key Laboratory of





Meteorological Disaster, Ministry of Education (No. KLME201810). This work is also supported by the Startup Foundation for introducing Talent of NUIST (Nos. 2015r002, 2014r011), China Postdoctoral Science Foundation Funded Project (No. 2016M591883), and Jiangsu Planned Projects for Postdoctoral Research Funds (No. 1601262C). We also gratefully appreciate the supports from Special Program for Applied Research on Super Computation of the NSFC-Guangdong Joint
Fund (the second phase) under Grant No. U1501501.

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

25

©c Author(s) 2019. CC BY 4.0 License.





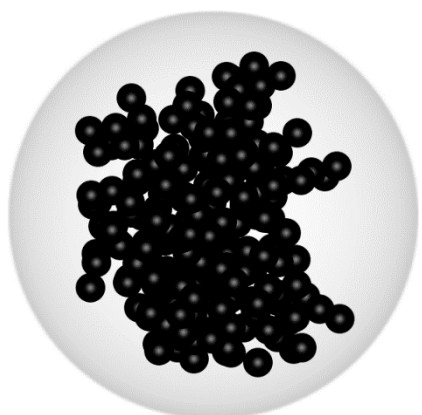

**Figure 1: Sketch map of geometry of coated black carbon. An example of fractal black carbon aggregates, containing 200 monomers, is coated by sulfate.**

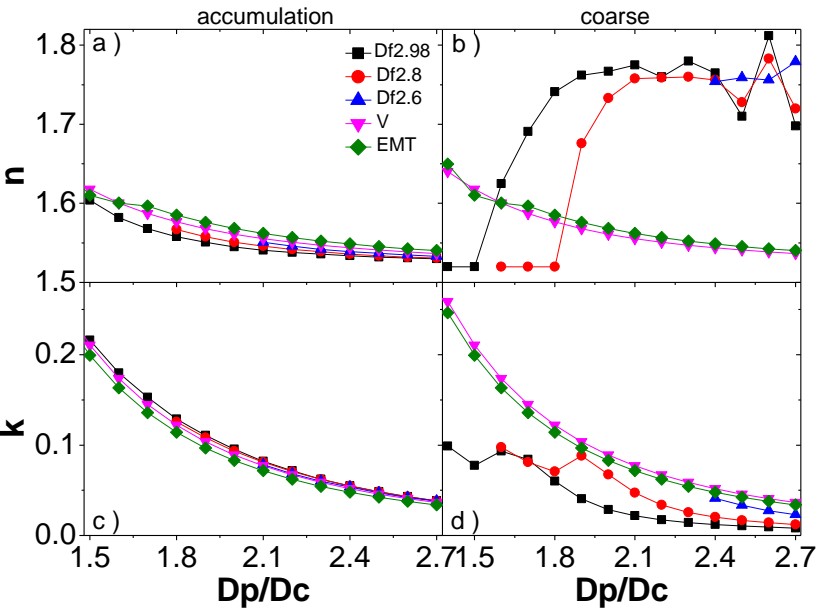

**Figure 2: The real (top row) and imaginary (bottom row) parts of retrieved optically effective aerosol complex refractive indices (ACRIs) of coated black carbon aggregates as a function of shell/core ratio (Dp/Dc, spherical volume-equivalent particle diameter/BC core diameter), in accumulation (left two) and coarse (right two) modes, respectively. Black squares, red circles and blue up-triangles indicate BC fractal dimensions of 2.98, 2.8 and 2.6, respectively. The ACRIs given by the popular volume weighted average method (magenta down-triangles) and Bruggeman effective medium theory (olive diamonds) are also considered for comparison.**





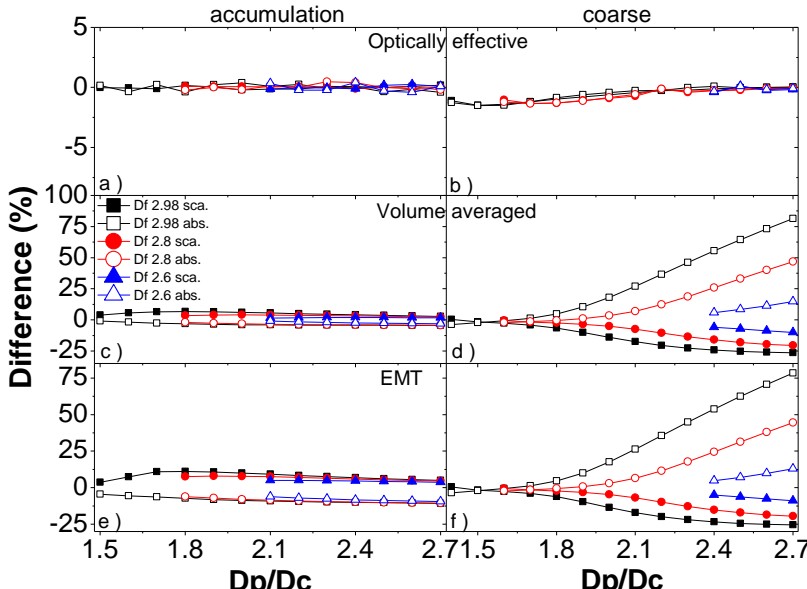

**Figure 3: The relative differences of scattering and absorption cross sections of coated black carbon aggregates induced by their optically effective complex refractive indices (top row), and refractive indices based on the volume weighted average method (middle row) and Bruggeman effective medium theory (bottom row) as a function of shell/core ratio (Dp/Dc), in accumulation (left three) and coarse (right three) modes, respectively. Black squares, red circles and blue triangles indicate BC fractal dimensions of 2.98, 2.8 and 2.6, respectively. Solid symbols denote scattering coefficients while open interiors indicate absorption coefficients.**





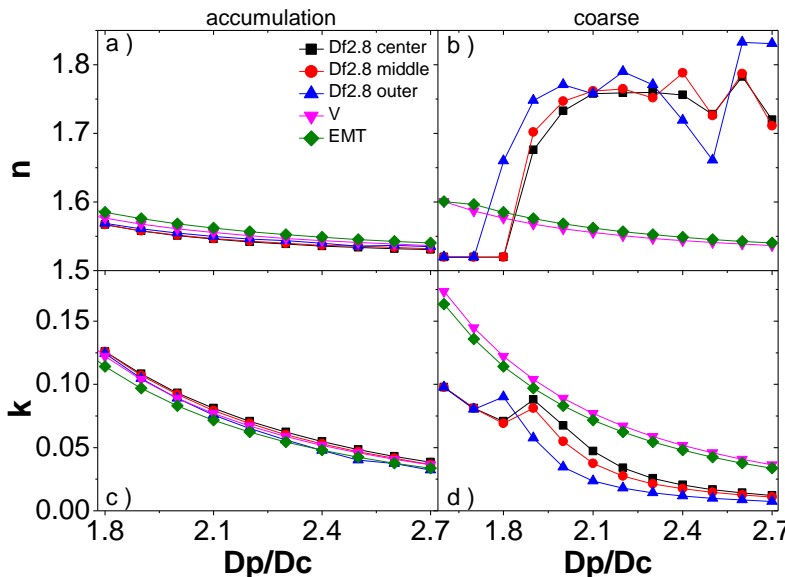

**Figure 4: The real (top row) and imaginary (bottom row) parts of retrieved optically effective aerosol complex refractive indices (ACRIs) of coated black carbon aggregates (BC fractal dimension of 2.8) as a function of shell/core ratio (Dp/Dc), in accumulation (left two) and coarse (right two) modes, respectively. Three BC coating morphologies are considered, i.e., a core-shell with BC mass center located at particle geometric center (black squares), and two off-center core-shell structures including BC aggregates lying at middle position of a particle radius (red circles) and outer position close to spherical boundary (blue up-triangles). The ACRIs based on the popular volume weighted average method (magenta down-triangles) and Bruggeman effective medium theory (olive diamonds) are compared.**



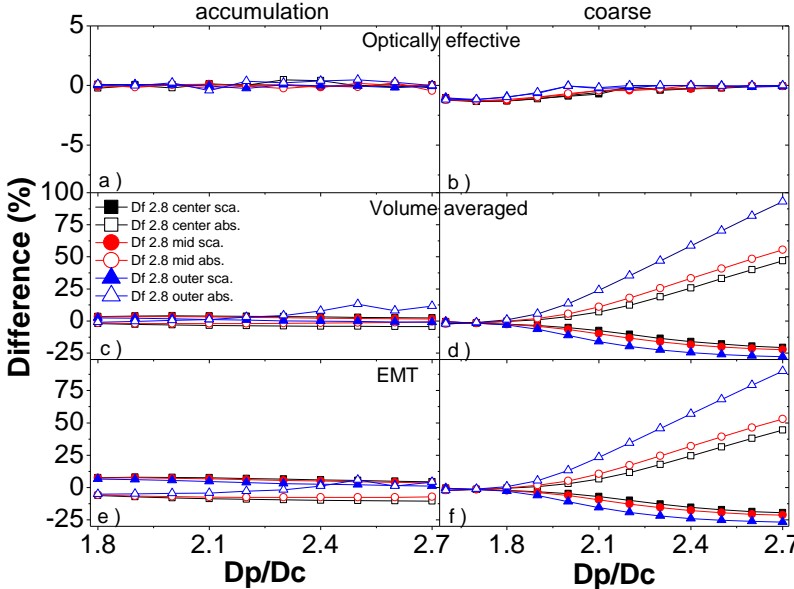

**Figure 5: The relative differences of scattering and absorption cross sections of coated black carbon aggregates (BC fractal dimension of 2.8) induced by their optically effective complex refractive indices (top row), and refractive indices based on the volume weighted average method (middle row) and Bruggeman effective medium theory (bottom row) as a function of shell/core ratio (Dp/Dc), in accumulation (left three) and coarse (right three) modes, respectively. Three BC coating morphologies are considered, i.e., a core-shell with BC mass center located at particle geometric center (black squares), and two off-center core-shell structures including BC aggregates lying at middle position of a particle radius (red circles) and outer position close to spherical boundary (blue triangles). Solid symbols denote scattering coefficients while open interiors indicate absorption coefficients.**



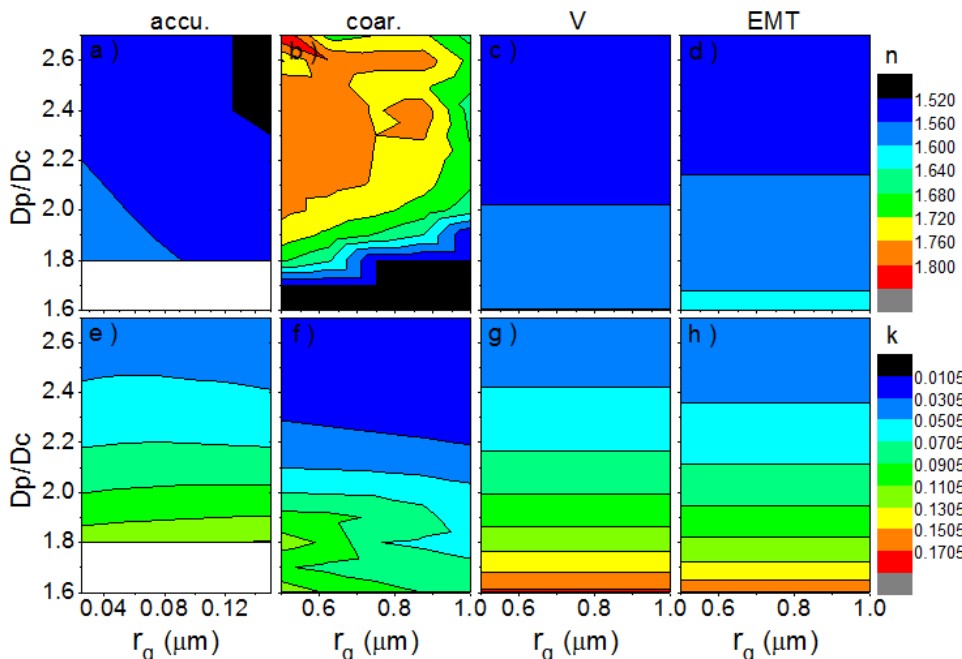

**Figure 6: The retrieved optically effective aerosol complex refractive indices of coated BC aggregates (BC fractal dimension of 2.8) with different shell/core ratio (Dp/Dc) and particle size distribution for accumulation (a, e) and coarse (b, f) modes, respectively. The estimated refractive indices given by the volume weighted average method (c, g) and Bruggeman effective medium theory (d, h) are considered for coarse mode. The real and imaginary parts correspond to top row and bottom row, respectively. The geometric standard deviations (σg) for applied lognormal distribution are 1.59 and 2.0 for accumulation and coarse aerosols, respectively.**





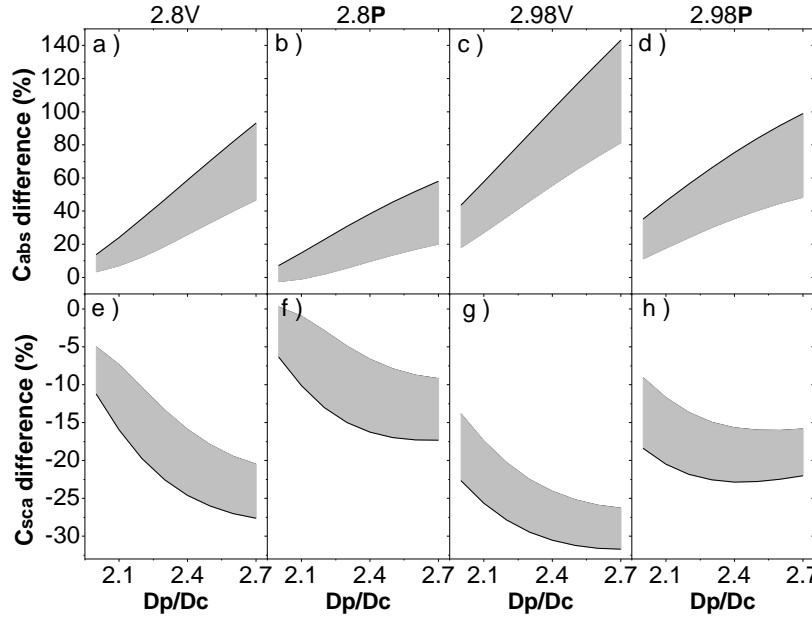

**Figure 7: Comparisons between the relative differences of scattering and absorption coefficients of coarse coated black carbon aggregates induced by the refractive indices based on the volume weighted average method (a, c, e, g) and those given by the parameterized method (b, d, f, h). Two BC fractal dimensions of 2.8 (left four) and 2.98 (right four) are considered, and the induce differences for scattering and absorption cross sections are shown in top and bottom rows, respectively. The solid lines indicate the cases of core-shell structures with BC mass center located at particle geometric center while the gray areas denote the cases of all possible core-shell structures with BC aggregates inside sulfate coating.**