# Peer review of "Optically effective complex refractive index of coated black carbon aerosols: from numerical aspects"

_Atmospheric Chemistry and Physics, 2018_

## Referee Comment (RC1) · Anonymous Referee #1 · 15 Apr 2019

Comments: The work by Zhang et al. investigated optically effective complex refractive index of coated BC by the multiple-sphere T-matrix method. They investigated the optically effective ACRI of polydisperse coated BC aggregates retrieved from their accurate scattering and absorption properties on different shell/core ratio, and compared with VWA and EMT. Besides, they propose a new ACRI parameterization for fully coated BC with $Dp/Dc \geq 2.0$ in coarse mode. The paper is overall well written. I suggested its publication in Atmospheric Chemistry and Physics after addressing the following issues: In line 8-10, the author said that "the simple spherical coatings on BC particles have similar effects on scattering and absorption properties to those with more complicated coating structures". But I didn't find the same conclusion from the

cited paper by Dong et al., [2015] and Liu et al., [2016], and in my opinion this point may be not true in reality. Please carefully reread these two papers. In line 19-21, more description about fractal dimension should be added, for example, wang et al., [2017] get a fractal dimension about 2.2 for aged BC aggregates. The size of BC in figure 2-5, 7 is not mentioned throughout the manuscript. This information should be added in Methods Section. In Methods Section, the method calculating the ACRIs of concentric coated BC aggregates with different particle size distributions are not mentioned. Although the author briefly introduced the method in Section 3.2, detailed methods and parameteres should be added in Methods Section. The sketch maps of geometry of coated black carbon of two off-center core-shell structures should also be shown in the manuscript or supplementary information section. In Section 3.3, the author needs to descript the exact size of BC aggregates. Besides, did you calculated different size and fractal dimension of BC aggregates using your new assumed parameterization of ACRI? Does this method always perform better? This should be illustrated. In section 3.4, Due to aged BC particles having complicated coating morphologies in ambient air. Seemly, individual particle analysis provide very good coating morphologies (Wang et al., ESTL, 2017, Adachi et al., JGR, 2008; Li et al., 2016, JGR)

References: Dong, J., J. Zhao, and L. H. Liu (2015), Morphological effects on the radiative properties of soot aerosols in different internally mixing states with sulfate, Journal of Quantitative Spectroscopy and Radiative Transfer, 165, 43-55, doi:10.1016/j.jqsrt.2015.06.025. Liu, F., J. Yon, and A. Bescond (2016), On the radiative properties of soot aggregates – Part 2: Effects of coating, Journal of Quantitative Spectroscopy and Radiative Transfer, 172, 134-145, doi:https://doi.org/10.1016/j.jqsrt.2015.08.005. Wang, Y., F. Liu, C. He, L. Bi, T. Cheng, Z. Wang, H. Zhang, X. Zhang, Z. Shi, and W. Li (2017), Fractal Dimensions and Mixing Structures of Soot Particles during Atmospheric Processing, Environmental Science & Technology Letters, 4(11), 487-493, doi:10.1021/acs.estlett.7b00418.

---

## Editor Comment (EC1) · Yves Balkanski (Editor) · 16 Apr 2019

This manuscript uses the multiple-sphere T-matrix method to evaluate the following effects on the refractive index of a mixture of black carbon with sulfate or with slightly absorbing aerosol. The effects are: shell/core geometry, BC position inside coating and size distribution. The obtained refractive indices are then compared to the ones calculated using simpler approximation such as the volume weighted averages or effective medium theory. The results shown are interesting and some of them are new. In one or two place the text would need to be clarified and some significant references are missing. I have tried to suggest to add these references where the reader would benefit

from knowing the work that has been done before. With some clarification and minor changes, this manuscript is suitable to be published in ACP after these modifications have been reviewed. Here are my suggesions:

Abstract lines 19-21 please simplify the following sentence that needs to be rewritten: ''However, in accumulation mode, the coated BC optically effective ACRI is dominantly influenced by particle chemical compositions and shell/core ratio, although it shows slightly sensitive to BC geometry, BC position inside coating and particle size distribution.''

Page 2 line 15: together with the reference to Shiraiwa et al 2010 you can add: Cui et al. 2016 and Peng et al 2016 (which is already in the ref. list)

Page 4 line 16: There is one '.' too many in '' based on Sorensen. [2001].''

Page 9, Section 3.2 When discussing the role of size distribution, you should mention the important work of Bond et al 2006 and in particular discuss your results in the light the Figure 4b of this paper, that will enrich this paragraph.

Page 9-10 Section 3.3 This is more than a standard sub-section, you propose here an approximation to correct for the erroneous values given by a simpler method such as Volume Weighted Average. You could indicate it in the sub-section title.

Page 11, I recommend that line 31 you replace the acronym MSTM with multi-sphere T-matrix method.

Page 12 line 24 replace "are still relatively poor" with "still show poor agreement"

Page 19 caption of Figure 3, please indicate that you mix BC with sulfate in the results of this Figure.

Figure 4: if these results are obtained with BC-sulfate mixing, please indicate it in the Fig. caption.

Page 22, Figure 6. How do you results compare with the ones of Fig. 4b from Bond et

al, JGR 2006?

Page 23 Figure 7; please indicate that the parameterized method is described by Eqs 7 & 8 in your main text.

References: 1. Cui X, et al. (2016) Radiative absorption enhancement from coatings on black carbon aerosols. Sci Total Environ 551-552:51–56. 2. Bond, T. C., G. Habib, and R. W. Bergstrom (2006), Limitations in the enhancement of visible light absorption due to mixing state, J. Geophys. Res., 111, D20211, doi:10.1029/2006JD007315.

---

## Author Comment (AC1) · 3 May 2019

Comments: The work by Zhang et al. investigated optically effective complex refractive index of coated BC by the multiple-sphere T-matrix method. They investigated the optically effective ACRI of polydisperse coated BC aggregates retrieved from their accurate scattering and absorption properties on different shell/core ratio, and compared with VWA and EMT. Besides, they propose a new ACRI parameterization for fully coated BC with $Dp/Dc \geq 2.0$ in coarse mode. The paper is overall well written. I suggested its publication in Atmospheric Chemistry and Physics after addressing the

following issues: In line 8-10, the author said that "the simple spherical coatings on BC particles have similar effects on scattering and absorption properties to those with more complicated coating structures". But I didn't find the same conclusion from the cited paper by Dong et al., [2015] and Liu et al., [2016], and in my opinion this point may be not true in reality. Please carefully reread these two papers.

√ This conclusion is drawn from Dong et al. [2015] and Liu et al. [2016] based on their comparisons of scattering and absorption properties of coated BC with more complicated structures with those of core-shell Mie model indirectly, and we have cited another paper (i.e., Liu et al., 2017) that gives this conclusion in its Abstract directly. (Page 4, line 6-8: "it is found that the simple spherical coatings on BC particles have similar effects on scattering and absorption properties to those with more complicated coating structures [e.g., Dong et al., 2015; F. Liu et al., 2015; C. Liu et al., 2017]")

In line 19-21, more description about fractal dimension should be added, for example, wang et al., [2017] get a fractal dimension about 2.2 for aged BC aggregates.

√ We have added some descriptions about BC fractal dimension accordingly. (Page 4, line 14-16: "The Df can characterize the shape of BC aggregates reasonably well, and its variation reflects BC aging processes [Wang et al., 2017]").

The size of BC in figure 2-5, 7 is not mentioned throughout the manuscript. This information should be added in Methods Section.

√ We have added accordingly. (Page 5, line 4-7: "For the accumulation mode, the radius range is set as 0.05–0.5 $\mu$m in steps of 0.005 $\mu$m, while the coarse radius range is assumed to be 0.5–2.5 $\mu$m in steps of 0.05 $\mu$m as ambient aerosols with size larger than 5 $\mu$m are few [Zhang et al., 2014, 2018; Zhang and Mao, 2015]. Note that the exact sizes of BC aggregates are known based on these coated BC sizes and shell/core ratios.")

In Methods Section, the method calculating the ACRIs of concentric coated BC aggre-

gates with different particle size distributions are not mentioned. Although the author briefly introduced the method in Section 3.2, detailed methods and parameters should be added in Methods Section.

√ We have revised accordingly. (Page 6, line 11-13: "As the optically effective ACRIs of coated BC with fixed microphysical parameters (such as shell/core ratio, BC fractal dimension, size distribution) are retrieved, it is possible to study the impacts of these microphysical parameters on retrieved optically effective ACRI with more details.")

The sketch maps of geometry of coated black carbon of two off-center core-shell structures should also be shown in the manuscript or supplementary information section.

√ We have added accordingly in the supplementary information section. (Fig. S1)

In Section 3.3, the author needs to descript the exact size of BC aggregates. Besides, did you calculated different size and fractal dimension of BC aggregates using your new assumed parameterization of ACRI? Does this method always perform better? This should be illustrated.

√ We consider the bulk particle scattering and absorption properties averaged over a certain size distribution, which is meaningful for ambient atmospheric applications. The exact sizes of coated BC aggregates considered are 0.1–1 $\mu$m in steps of 0.01 $\mu$m for accumulation mode, and 1–5 $\mu$m in steps of 0.1 $\mu$m for coarse mode. Then the exact size of BC is restricted by shell/core ratio based on these coated BC sizes. We have added the size distribution assumed in Section 3.3 (page 10, line 21). For the performance of our new assumed parameterization of ACRI, BC fractal dimensions of 2.8 and 2.98 have already been considered in the manuscript. For monodisperse particle size, Fig. S2 compares induced relative errors of scattering and absorption coefficients of coarse coated BC at different size with BC fractal dimension of 2.8 and shell/core ratio of 2.7 based on ACRIs from the popular VWA and Equations (7-8). The results show that compared to the VWA, our simple new parameterization method reduces the relative errors in estimating absorption cross sections of coarse coated

**[ACPD](https://www.atmospheric-chemistry-and-physics.net/)**
BC at all particle size selected. Meanwhile, the relative errors of coated BC scattering cross are lessened at almost all monodisperse size, and this may be due to that our simple method only considers the correction of imaginary part of ACRI and its real part is the same as that based on the VWA. However, with the size distribution considered, the errors of both absorption and scattering cross sections of coarse coated BC are reduced on the basis of our new assumed parameterization method in comparison of the VWA. (Fig. S2)

In section 3.4, Due to aged BC particles having complicated coating morphologies in ambient air. Seemly, individual particle analysis provide very good coating morphologies (Wang et al., ESTL, 2017, Adachi et al., JGR, 2008; Li et al., 2016, JGR) References: Dong, J., J. Zhao, and L. H. Liu (2015), Morphological effects on the radiative properties of soot aerosols in different internally mixing states with sulfate, Journal of Quantitative Spectroscopy and Radiative Transfer, 165, 43-55, doi:10.1016/j.jqsrt.2015.06.025. Liu, F., J. Yon, and A. Bescond (2016), On the radiative properties of soot aggregates – Part 2: Effects of coating, Journal of Quantitative Spectroscopy and Radiative Transfer, 172, 134-145, doi:https://doi.org/10.1016/j.jqsrt.2015.08.005. Wang, Y., F. Liu, C. He, L. Bi, T. Cheng, Z. Wang, H. Zhang, X. Zhang, Z. Shi, and W. Li (2017), Fractal Dimensions and Mixing Structures of Soot Particles during Atmospheric Processing, Environmental Science & Technology Letters, 4(11), 487-493, doi:10.1021/acs.estlett.7b00418.

√ We have revised it accordingly. (page 11, line 7-8: "Due to aged BC particles having complicated coating morphologies in ambient air, which can be provided by individual particle analysis [Adachi and Buseck, 2008; Li et al., 2016; Wang et al., 2017],")

Please also note the supplement to this comment:
https://www.atmos-chem-phys-discuss.net/acp-2018-1279/acp-2018-1279-AC1-supplement.zip

---

## Author Comment (AC2) · 3 May 2019

Yves Balkanski (Editor) yves.balkanski@lsce.ipsl.fr

This manuscript uses the multiple-sphere T-matrix method to evaluate the following effects on the refractive index of a mixture of black carbon with sulfate or with slightly absorbing aerosol. The effects are: shell/core geometry, BC position inside coating and size distribution. The obtained refractive indices are then compared to the ones calculated using simpler approximation such as the volume weighted averages or effective medium theory. The results shown are interesting and some of them are new. In one or two place the text would need to be clarified and some significant references are

missing. I have tried to suggest to add these references where the reader would benefit from knowing the work that has been done before. With some clarification and minor changes, this manuscript is suitable to be published in ACP after these modifications have been reviewed. Here are my suggestions: Abstract lines 19-21 please simplify the following sentence that needs to be rewritten: "However, in accumulation mode, the coated BC optically effective ACRI is dominantly influenced by particle chemical compositions and shell/core ratio, although it shows slightly sensitive to BC geometry, BC position inside coating and particle size distribution."

√ We have revised accordingly. (Page 1, line 19-20: "However, in accumulation mode, the coated BC optically effective ACRI is dominantly influenced by particle chemical compositions and shell/core ratio.")

Page 2 line 15: together with the reference to Shiraiwa et al 2010 you can add: Cui et al. 2016 and Peng et al 2016 (which is already in the ref. list)

√ We have revised it accordingly. (Page 2, line 13)

Page 4 line 16: There is one '.' too many in '' based on Sorensen. [2001]."

√ We have revised accordingly. (Page 4, line 14: "The kf of BC is assumed to be 1.2 based on Sorensen [2001].")

Page 9, Section 3.2 When discussing the role of size distribution, you should mention the important work of Bond et al 2006 and in particular discuss your results in the light the Figure 4b of this paper, that will enrich this paragraph.

√ We have mentioned this important work of Bond et al. [2006]. (Page 9, line 4-5: "As demonstrated in Bond et al. [2006], particle size distribution affects coated BC absorption properties and its BC absorption amplification due to weakly absorbing coatings." Page 9, line 23-25: "This is consistent with the results of Bond et al. [2006], which suggest that the VWA for refractive index is unrealistic, leading to unphysical results and overestimating particle absorption.")

Page 9-10 Section 3.3 This is more than a standard sub-section, you propose here an approximation to correct for the erroneous values given by a simpler method such as Volume Weighted Average. You could indicate it in the sub-section title.

√ We have revised accordingly. (Page 9, line 26: "3.3 A new assumed ACRI parameterization for heavily coated BC as a correction for the VWA approximation")

Page 11, I recommend that line 31 you replace the acronym MSTM with multi-sphere T-matrix method.

√ We have revised it accordingly. (Page 12, line 4)

Page 12 line 24 replace "are still relatively poor" with "still show poor agreement"

√ We have revised accordingly. (Page 12, line 27).

Page 19 caption of Figure 3, please indicate that you mix BC with sulfate in the results of this Figure.

√ We have revised accordingly. (Figure 3)

Figure 4: if these results are obtained with BC-sulfate mixing, please indicate it in the Fig. caption.

√ We have revised it accordingly. (Figure 4)

Page 22, Figure 6. How do you results compare with the ones of Fig. 4b from Bond et al, JGR 2006?

√ We thank the suggestion from the reviewer, and have read this famous paper carefully. The Fig. 4b of Bond et al. (JGR, 2006) depicts the absorption amplification of BC due to coating at different shell-core size regions, whereas the Fig. 6 of our manuscript shows retrieved optically effective ACRIs at different size distributions and shell/core ratios. Nevertheless, we have mentioned this famous work of Bond et al. (JGR, 2006), and compared the VWA results for refractive index. (Page 9, line 4-5: "As demonstrated

in Bond et al. [2006], particle size distribution affects coated BC absorption properties and its BC absorption amplification due to weakly absorbing coatings." Page 9, line 23-25: "This is consistent with the results of Bond et al. [2006], which suggest that the VWA for refractive index is unrealistic, leading to unphysical results and overestimating particle absorption.")

Page 23 Figure 7; please indicate that the parameterized method is described by Eqs 7 & 8 in your main text. References: 1. Cui X, et al. (2016) Radiative absorption enhancement from coatings on black carbon aerosols. Sci Total Environ 551-552:51–56. 2. Bond, T. C., G. Habib, and R. W. Bergstrom (2006), Limitations in the enhancement of visible light absorption due to mixing state, J. Geophys. Res., 111, D20211, doi:10.1029/2006JD007315.

√ We have modified it accordingly. (Figure 7)

Please also note the supplement to this comment:
https://www.atmos-chem-phys-discuss.net/acp-2018-1279/acp-2018-1279-AC2-supplement.zip
* * *